# Hot Deformation Characteristics and 3-D Processing Map of a High-Titanium Nb-Micro-alloyed Steel

**DOI:** 10.3390/ma13071501

**Published:** 2020-03-25

**Authors:** Pingping Qian, Zhenghua Tang, Li Wang, Charles W. Siyasiya

**Affiliations:** 1Department of Materials Science and Engineering, Sichuan University, Chengdu 610000, China; ppqiansc@outlook.com (P.Q.); liwangsc@outlook.com (L.W.); 2Department of Materials Science and Metallurgical Engineering, University of Pretoria, Tshwane 999136, South Africa; Siyasiya@126.com

**Keywords:** high-titanium Nb-micro-alloyed steel, deformation behavior, constitutive modelling, traditional and 3-D processing maps

## Abstract

Hot deformation behavior of a high-titanium Nb-micro-alloyed steel was investigated by conducting hot compression tests at the temperature of 900–1100 °C and the strain rate of 0.005–10 s^−1^. Using a sinh type constitutive equation, the apparent activation energy of the examined steel was 373.16 kJ/mol and the stress exponent was 6.059. The relations between Zener–Hollomon parameters versus peak stress (strain) or steady-state stress (strain) were successfully established via the Avrami equation. The dynamic recrystallization kinetics model of the examined steel was constructed and the validity was confirmed based on the experimental results. The 3-D atomic distribution maps illustrated that strain can significantly affect the values of power dissipation efficiency and the area of instability domains. The 3-D processing maps based on a dynamic material model at the strains of 0.2, 0.4, 0.6 and 0.8 were established. Based on traditional and 3-D processing maps and microstructural evaluation, the optimum parameter of for a high-titanium Nb-micro-alloyed steel was determined to be 1000–1050 °C/0.1–1 s^−1^.

## 1. Introduction

Hot deformation behavior of high-strength low-alloy steels has become the focus of many researchers for many years because of higher yield strength, lower impact transition temperature and better cold formability. The micro-alloyed steel has wide application potential in diverse fields such as petroleum pipelines, automobiles and rail transits [1,2]. However, the defects of casting products, such as porosity and shrinkage, mean that they cannot meet the requirements of high-performance structural materials, greatly limiting their application. The hot deformation of micro-alloyed steels resulting in grain refinement improves the strength and toughness of materials and eliminates various defects. For micro-alloyed steels, the entire hot rolling process is often accompanied by solid solution and late precipitation due to the addition of micro-alloy elements such as Nb, V, Ti and N etc [3,4]. This makes recrystallization of austenite more complicated and difficult to predict. The constitutive equation can be used to describe the variation of flow stress under different hot processing parameters, such as deformation temperature, strain and strain rate, which is a good tool to determine the optimal processing parameter [5,6,7].

It is an effective method to utilize a processing map to design hot processing parameters and improve the properties of materials. Raj et al. proposed to build a processing map on the basis of cavity nucleation mechanisms, dynamic recrystallization and adiabatic heating effects [8]. This processing map, however, has limitations that it can only be applied to pure metals and simple alloys, but not to the deformation mechanism of complex alloys. Based on the dynamic material model (DMM), Gegel et al. developed a processing map which requires a superimposition of the power dissipation efficiency map and the instability map in the frame of the logarithm of strain rate and deformation temperature [9]. The approach of the dynamic material model has been widely used in various alloys, such as Ni base superalloys [10,11], steels [12,13,14,15], Ti alloys [16,17], Zr alloys [18], Mg alloys [19,20,21,22] and high-entropy alloys [23]. The internal workability for these materials is discussed through the establishment of a processing map. It is an effective supplementary technique to be used for optimizing the hot processing parameters.

Although some studies have studied the hot deformation behavior of Nb–Ti micro-alloyed steel [24], few researchers have studied the effect of 3-D processing maps considering strain and high-titanium on the hot deformation behavior of Nb steel. In the present work, the hot deformation behavior of a high-titanium Nb-micro-alloyed steel is investigated using hot compression tests under different strain rates and deformation temperatures. In addition, traditional and 3-D processing maps considering strain are developed to optimize hot processing parameters and verify its accuracy on the basis of microstructural evaluation. The equations of flow models such as stress exponent, Zener-Hollomon, apparent activation energy, dynamic recrystallization fraction and 3-D processing maps of the material are established to optimize the hot processing parameters.

## 2. Examined Steel and Procedures

The examined steel used in this study is an Nb–Ti micro-alloyed steel and its chemical composition is given in Table 1. The test samples were processed into a cylindrical specimen with a diameter of 10 mm and height of 15 mm and it was then subjected to a single pass hot compression test in a thermal simulator (model Gleeble 3500).

In order to prevent oxidation during hot deformation, high-purity Ar was used as protective gas, and graphite powder was coated on both ends of the specimens to reduce the bulging effect caused by end friction. The specimens were reheated to 1200 °C at a rate of 10 °C/s, held for 5 min followed by cooling down to testing temperature (900, 950, 1000, 1050 and 1100 °C). Before hot compression, the specimens were held at the deformation temperature for 2 min to eliminate the temperature gradient. The Single-pass compression tests were carried out at different strain rates (0.005, 0.1, 1 and 10 s^−1^) up to a true strain of 0.8. The specimens were then water quenched immediately after compression to preserve the deformed microstructure. The true stress-strain curve was obtained by transforming the load-stroke data during hot compression [25].

The deformed specimens were sectioned parallel to the compression axis for microstructural observation. The specimens were mechanical polished and etched in an abluent solution of saturated picric acid (15 mL picric acid, 200 mL water and 0.15 mL of hydrofluoric acid). The microstructures of the samples were observed using a light microscope (LM) (KH-2200MD2, Tokyo, Japan). The deformation dislocation was observed using a transmission electron microscopy (TEM) (model JEM-2100F, Tokyo, Japan). The samples were cut to about 400 μm thickness and thinned to 40–50 μm by mechanical polishing. The thinned samples were then subjected to a double-jet polisher (Tenupol-5, Copenhagen, Denmark) in perchloric acid solution to reach the final thickness.

## 3. Results and Discussion

### 3.1. True Stress–True Strain Curves

The true stress-true strain curves of the examined steel at different deformation temperatures for a given strain rate were analyzed (Figure 1). When the strain rate is 0.1 s^−1^ and the deformation temperature is higher than 950 °C (Figure 1b), the flow stress decreases after reaching a peak stress value and then stays at a constant stress value, which indicates that the phenomenon of dynamic recrystallization (DRX) occurs. This is because the softening effect reduces the dislocation density, which leads to the decrease of flow stress. When the deformation temperature is less than 950 °C (Figure 1c), the flow stress keeps rising and then stabilizes with the increase of strain, which means that the phenomenon of dynamic recovery (DRV) occurs. It shows that the work hardening of the examined steel has played a dominant role in the hot deformation process. In short, the flow stress of materials in the process of hot deformation is affected by various factors. 

The relation between peak flow stress with strain rates and deformation temperatures at a true strain of 0.8 was analyzed (Figure 2). The peak flow stress increased as the increasing of strain rate at a certain deformation temperature and the peak flow stress gradually decreases as the deformation temperature increases. At high temperatures, the energy barrier of dislocation motion decreases due to the hot activation, then the dynamic restoration is accelerated and flow stress decreases.

### 3.2. Constitutive Analysis

The deformation behavior of materials at high temperature can be expressed by constitutive equation, which describes the relations among strain rate, deformation temperature and flow stress, and is represented by Equation (1) [6].
(1)$$Z=\dot{\varepsilon }\exp\left(-\frac{Q_{def}}{RT}\right)=\left\{ \begin{array}{l} A_{1}{\left(\sigma \right)}^{n^{\prime }}(\alpha \sigma <0.8)\\ A_{2}\exp(\beta \sigma )(\alpha \sigma >1.2)\\ A{\left[\sinh(\alpha \sigma )\right]}^{n}(\mathrm{for}\;\mathrm{all}\;\sigma ) \end{array}\right.$$
where *Q_def_* is the hot deformation activation energy (KJ/mol), σ is the strain rate (s^−1^), *T* is the absolute temperature (K), *R* is the universal gas constant, and finally *A*_1_, *A*_2_, *A*, *α*, *β*, *n* and *n*′ are the material constants.

According to Equation (1), the partial differentiation of the power and exponential laws at a given deformation temperature are as follows:(2)$$n^{\prime }={\left[\partial \ln\dot{\varepsilon }/\partial \ln\sigma_{p}\right]}_{T}$$
(3)$$\beta ={\left[\partial \ln\dot{\varepsilon }/\partial \sigma_{p}\right]}_{T}$$


The values of the above expressions *n*′ and *β* can be obtained from the slopes of the curves of $\ln\dot{\varepsilon }$ vs. ln*σ_p_* and $\ln\dot{\varepsilon }$ vs. *σ_p_*, respectively. As shown in Figure 3a,b, the average slopes of these curves are 8.219 and 0.0728 for *n*′ and *β*, respectively. Therefore, the value of *α* = *β*/*n*′ is determined as 0.0089.

At the same temperature, the plot of ln[sinh(*ασ_p_*)] vs. $\ln\dot{\varepsilon }\;$atisfies the linear relationship, as shown in Figure 3c. This illustrates that the stress index is independent of temperature and the average value of n is 6.059. The plot of ln[sinh(*ασ_p_*)] vs. 1/*T* (as shown in Figure 4) also shows a linear relationship, and the average slope b is determined to be 7407.25. According to the intercept of fitting line in Figure 5, the value of A is 1.618 × 10^14^, then the hyperbolic sine constitutive model of the examined steel at a true strain of 0.8 can be expressed as follows:(4)$$Z=\dot{\varepsilon }\exp(373.16/RT)=1.618\times 10^{14}{\left[\sinh(0.0089\sigma_{p})\right]}^{6.059}$$

This gives the value *Q_def_* = *Rnb* = 373.16 KJ/mol, which is consistent with similar results reported in the literature [26]. The value of *Q_def_* and the stress index *n* of the examined steel are higher than Ti–Nb steels tested with low titanium and low niobium content [27,28]. The values around 290 and 305 KJ/mol seem to be reasonable for activation energies for hot deformation in the case of Ti IF and Ti Nb IF steels [29]. The addition of micro-alloying elements Nb and Ti it is known to delaying recovery and preventing static recrystallization to commence, leading to a strain accumulation that can be caused or by fine particle precipitation in austenite.

### 3.3. Characterization of DRX Behavior

The DRX is a critical softening process for examined steel in the hot deformation behavior. The strain hardening rate *θ* (*θ* = d*σ*/d*ε*) can be used to better understand the change process of stress in the hot deformation. The variation of strain hardening rate with stress at the strain rate of 1 s^−1^ are given in Figure 6. It can be seen from the curve that as the stress increases, the strain hardening rate gradually decreases until the stress reaches the peak value, at which time the strain hardening rate drops to zero. According to Jonas et al. [30], strain hardening rate increases again as the stress decreasing, and when it reaches zero again, the stress reaches the steady-state stress (*σ_s_*). 

Based on the power functions of *Z* = *A*′*σ^n^*_1_ and *Z* = *A*″*ε^n^*_2_, the relations between the *Z* parameter and *σ*(*ε*) can be obtained. According to the above equation, the plots of ln*σ_p_*, ln*ε_p_*, ln*σ_s_* and ln*ε_s_* vs. ln*Z* are shown in Figure 7. Therefore, the relationships between peak strain or stress and steady-state strain or stress with the *Z* parameter are described as follows:(5)$$\sigma_{p}=1.67Z^{0.12}$$
(6)$$\sigma_{s}=0.768Z^{0.15}$$
(7)$$\varepsilon_{p}=0.0028Z^{0.13}$$
(8)$$\varepsilon_{s}=0.142Z^{0.05}$$
where the subscripts s and p represent steady-state and peak, respectively. Among them Equations (5)–(8) describe the direct relations between the characteristic stress (strain) and *Z* parameter.

According to Cho et al., the dynamic recrystallization fraction can be expressed in the form of the Avrami equation as follows [31]:(9)$$X=1-\exp[-k{\left((\varepsilon -\varepsilon_{c})/\varepsilon_{p}\right)}^{n_{3}}]$$
where *k* and *n*_3_ are the material constants. The maximum softening rate strain, *ε*^∗^ and the strain required to achieve steady-state stress, *ε_s_*, are important parameters to characterize the volume fraction of DRX in hot deformation. According to some studies, it is proven that Equation (2) can be transformed as follows [32]:(10)$$X=(\sigma_{p}-\sigma )/(\sigma_{p}-\sigma_{s})=1-\exp[-k(2(\varepsilon -\varepsilon_{p})/{\left(\varepsilon_{s}+\varepsilon_{p}\right)}^{n_{3}}]$$

By logarithmic processing on both sides of Equation (10), it can be seen from Figure 8 that the relationship between lnln(1/(1 − *X*)) and ln(2(*ε* − *ε_p_*)/(*ε_s_* + *ε_p_*)) is almost linear at different deformation temperatures. The slope and intercept of the fitted linear equation (Figure 8) correspond to the values of the material coefficients k and n_3_, that is, they are 1.4847 and 3.4931, respectively. Therefore, the Avrami equation for dynamic recrystallization of examined steel is expressed as follows:(11)$$X=1-\exp[-3.4931(2(\varepsilon -\varepsilon_{p})/{\left(\varepsilon_{s}+\varepsilon_{p}\right)}^{1.4847}]$$

The dynamic recrystallization based on the Avrami equation is shown in Figure 9. It can be seen that all the curves exhibit “*s*” shapes, which indicates that the volume fraction of DRX increases slowly as the increase of strain at the beginning of deformation, sharply in the middle and then slowly near the end of deformation. At the same strain rate, the curve shifts to the left as the temperature increases. Compared with low titanium and low niobium micro-alloyed steel, higher titanium and niobium elements are beneficial to obtain a higher n_3_ value. This is because the *n*_3_ value depends on the nucleation mechanism of recrystallization and type of nucleation [33,34]. Most of the nucleation occurs in the early stage of recrystallization, and the process is easy to occur near the grain boundary, forming a so-called necklace structure after a short time [32]. In this process, high strain rate and low deformation temperature will significantly increase the starting point of DRX, that is, the critical strain value. The dynamic model of this experiment can predict the volume fraction of DRX of a high-titanium Nb-micro-alloyed steel very well.

### 3.4. Processing Maps

#### 3.4.1. The Principles of Processing Maps

The dynamic material model is proposed to reveal the relationship between material flow behavior and microstructural evolution. Processing maps are designed to avoid defects and flow instability areas, and obtain an optimum hot workability in the process of material deformation. According to the theory of dissipative structure, the energy P of input system can be divided into two parts: content (*G*) and co-content (*J*), which are mathematically defined as [35]:(12)$$P=\sigma \cdot \dot{\varepsilon }=G+J=\int_{0}^{\dot{\varepsilon }}\sigma \cdot d\dot{\varepsilon }+\int_{0}^{\sigma }\dot{\varepsilon }\cdot d\sigma$$
where *G* is the energy consumed by the plastic deformation of the material, and *J* is the energy consumed by the evolution of the microstructure in the plastic deformation of materials. 

In the process of plastic deformation, most of the energy will be converted into heat and released, and a small part of the energy will be dissipated in microstructural evolution. The strain rate sensitivity index m is used to determine the distribution relationship between *G* and *J* [36].
(13)$$m=\frac{\partial J}{\partial G}=\frac{\dot{\varepsilon }\partial \sigma }{\sigma \partial \dot{\varepsilon }}=\frac{\partial \ln\sigma }{\partial \ln\dot{\varepsilon }}$$

Strain rate sensitivity m is an important factor in describing the hot workability of materials. 

The efficiency of power dissipation η is introduced to represent the ratio of *J* and *J*_max_ (*m* = 1) of the ideal linear condition, and is defined as follows [36]:(14)$$\eta =\frac{J}{J_{\max}}=\frac{2J}{P}=\frac{2m}{m+1}$$

For an ideal dissipator, which is also known a linear dissipator, *m* = 1 and *G* = *J*. The power dissipation efficiency *η* is used to evaluate the power-dissipation capacity of the material. It can be stated that the optimum hot working condition is when *η* is maximum.

The flow instability parameter *ξ* is proposed according to the principle of maximum entropy production rate and used to determine flow instability of materials [9].
(15)$$\xi (\dot{\varepsilon })=\frac{\partial \log\left(m/(1+m)\right)}{\partial \log\dot{\varepsilon }}+m$$

The flow instability occurs in the system when the flow instability parameter *ξ* is less than zero.

#### 3.4.2. 3-D Distribution Maps of the Power Dissipation Efficiency and Instability Parameter under Different Conditions

In order to better understand the variation of processing parameters, a 3-D distribution map was established to clarify the relationship between hot working parameters and power dissipation efficiency and instability criteria. The variation of the power dissipation efficiency η under different deformation conditions is displayed in Figure 10a. Viewed as a whole, the value of *η* is mainly concentrated in the range of 0.15–0.3. In addition, when the strain is at 0.2 and 0.4, the values of *η* fluctuates less with temperature and strain rate. However, as the strain increases, the values of *η* shows a large fluctuation, which indicates that the strain rate and temperature at a large strain significantly affect the value of power dissipation efficiency of the materials. The instability parameters map (as shown in Figure 10b) considering the influence of strains is constructed to identify the variation of the instability parameter under different temperature and strain rate. The instability parameter *ξ* vary with strains and its value mainly fluctuates near *ξ* = 0. Evidently, as the strain increases, the value of the instability parameter of the materials fluctuates significantly, which appears as the same trend as the power dissipation efficiency. When the strain is at 0.8, serious instability occurs at a deformation temperature of 950 °C and strain rates of 10 s^−1^, and the value of instability parameter is −3.65.

The 3-D (three-dimensional) distribution maps (Figure 11) considering the influence of strain are established to determine optimum processing parameters. The contours of different colors mean the value of the power dissipation efficiency with varying temperature, strain rate and strains in the 3-D power dissipation map (Figure 11a). It can be seen from the figure that as the strain increases, the value of power dissipation efficiency increases sharply at the beginning and reaches a maximum at a strain of 0.6, and then a significant decrease occurs. This is because the deformation storage energy increases with the increase of strain, making the dynamic recrystallization grains easier to nucleate and grow. At the same time, the maximum value of η in the 3-D power dissipation map is about 0.34, and the higher values are mainly located at 0.1–1 s^−1^ and 1000–1100 °C. This phenomenon also validates the results of the latter observation of microstructures. The 3-D distribution map (Figure 11b) of instability parameter under strains of 0.2, 0.4, 0.6 and 0.8 was analyzed. The red area displays “instability” while the blue area mean “stability”. On the whole, as the strain increases, the area of the instability domains decreases rapidly and then increases, which is consistent with the trend of the 3-D power dissipation map. When the strain is at 0.6, the area of the instability domains is the smallest, and the instability domains are mainly located at 900–1100 °C/1–10 s^−1^. Comparing strain of 0.4 with strain of 0.2, it was found that the unsafe domains disappeared in the range of low temperature and strain rate. In addition, it is always safe in the region of moderate deformation temperature and strain rate under all strains.

#### 3.4.3. Processing Maps Analysis

The processing maps of Nb–Ti micro-alloyed steel obtained by superposition of the PDM (power dissipation map) and the instability map under true strains of 0.6 and 0.8 were displayed in Figure 12. The area of the instability domains with the increase of strain leads to a new unsafe domain. In terms of values of power dissipation efficiency and instability parameter, all strains exhibit similar distribution characteristics. Combined with processing maps analysis, the optimum designing range, which gives the highest values of power dissipation efficiency and is also in the safe area, can be determined as 1000–1050 °C/0.1–1 s^−1^. Therefore, processing maps can be effectively used to evaluate the optimum processing parameters of materials [37]. However, it is necessary to discuss the relationship between the processing maps and the microstructural evolution, and it can well verify the accuracy of the prediction of the processing maps.

#### 3.4.4. Microstructural Observations

The dynamic recrystallization easily occurs near the grain boundaries during hot deformation as shown in Figure 13. The original austenite grains are extruded into a flat shape, and fine equiaxed grains are created near the grain boundaries. The dislocation density near the grain boundary of the deformation band is higher than that in the interior of grains, which will facilitate nucleation along the grain boundary. When the examined steel deforms in the flow instability domains, it will incur inhomogeneous microstructure (Figure 13b). Therefore, the flow instability domains should be avoided in the hot deformation process of materials, which is conducive to obtaining uniform and fine grains and reducing internal defects. The typical microstructures of specimens deformed in instability and stability regions were characterized, as shown in Figure 14. As depicted in Figure 14a,b, at a fixed deformation temperature, the strain rate significantly affects the variation of grain size. The coarse grains are obtained at a high strain rate because the high strain rate increases the work hardening rate and shortens the austenite recrystallization time and grain growth time. The influence of hot deformation temperature on microstructure evolution is shown in Figure 14c,d. With the increase of temperature, the recrystallized grains have enough time to grow at low strain rate, leading to the coarse grains seen at 1050 °C. This is because higher temperatures can provide sufficient energy for the nucleation of DRX, but too high energy causes abnormal growth of grains. On the other hand, the microstructures also verify that the high-power dissipation efficiency in the 3-D power dissipation maps is mainly distributed in this safe domain. According to the microstructural evolution and traditional and 3-D processing maps, it is determined that the optimum hot deformation parameters of a high-titanium Nb-micro-alloyed steel is 1000–1050 °C/0.1–1 s^−1^.

The TEM photographs of samples after hot deformation at 1050 °C/0.1 s^−1^ and 1050 °C/10 s^−1^ were analyzed (Figure 15), in which a high density of dislocations was observed with a high magnification. When the deformation temperature decreases and strain rate increases, dislocations continue to occur through the source of dislocations and move to the grain boundary by sliding and climbing, which leads to the increase of dislocation density and the formation of a dislocation wall. The dislocation density near the grain boundary is higher, and the dislocation pile-up group is more obvious at high strain rates, as shown in Figure 15. This is because the dislocation pile-up near grain boundaries at high strain rates does not have enough time to be eliminated by the dynamic recovery and recrystallization. Combining Figure 11a and Figure 15, it is found that the values of power dissipation efficiency of the material are inversely related to the dislocation density. For example, the dislocation density is higher at 1050 °C/10 s^−1^, but the corresponding power dissipation efficiency is only 7.2%. When the strain rate drops to 0.1 s^−1^, the values of power dissipation efficiency at this time rises to 30.5%. Meanwhile, it also explains that the 3-D processing map can guide the evolution process of the microstructure.

## 4. Conclusions

The hot deformation behavior of a high-titanium Nb-micro-alloyed steel was analyzed by s constitutive equation, traditional and 3-D processing maps and microstructural evolution. The major results of this research can be summarized as follows:(1)Both the peak stress and flow stress increased with decreasing deformation temperature and increasing strain rate. The peak stress of a high-titanium Nb-micro-alloyed steel varied from the highest value of 219.7 MPa at 900 °C with a strain rate of 10 s^−1^ to the lowest value of 45.4 MPa at 1100 °C with a strain rate of 0.005 s^−1^.(2)Increasing the titanium and niobium content of micro-alloyed steel can increase the activation energy of hot deformation, which may hinder the occurrence of dynamic recrystallization. For high-titanium Nb-micro-alloyed steel, the constitutive equation was determined as(16)$$Z=\dot{\varepsilon }\exp(373.16/RT)=1.618\times 10^{14}{\left[\sinh(0.0089\sigma_{p})\right]}^{6.059}$$(3)The relations between the Zener–Hollomon parameters versus peak stress (strain) or steady-state stress (strain) were established via the Avrami equation. The DRX kinetics model of the examined steel was obtained as $X=1-\exp[-3.4931(2(\varepsilon -\varepsilon_{p})/{\left(\varepsilon_{s}+\varepsilon_{p}\right)}^{1.4847}]$. The high strain rate and low deformation temperature can significantly increase the starting point of DRX, that is, the critical strain value. A high agreement was recognizable and the kinetics model was suitable to predict the DRX process of the examined steel.(4)The power dissipation efficiency and instability maps based on the dynamic material model at the strains of 0.2, 0.4, 0.6 and 0.8 were established. The value of power dissipation efficiency and the area of instability domains vary with the increase of strain. The high η-value domain appears at 1000–1050 °C/0.1–1 s^−1^, and the instability domains occurred mainly in the high strain rate region (1–10 s^−1^) and low deformation temperature and strain rate region (900–950 °C/0.005–0.1 s^−1^). Based on the traditional and 3-D processing maps and microstructural evaluation, the optimum parameter of for a high-titanium Nb-micro-alloyed steel was at 1000–1050 °C/0.1–1 s^−1^, and a dynamic recrystallisation structure with fine and homogeneous grain size can be obtained.

## Figures and Tables

**Figure 1 materials-13-01501-f001:**
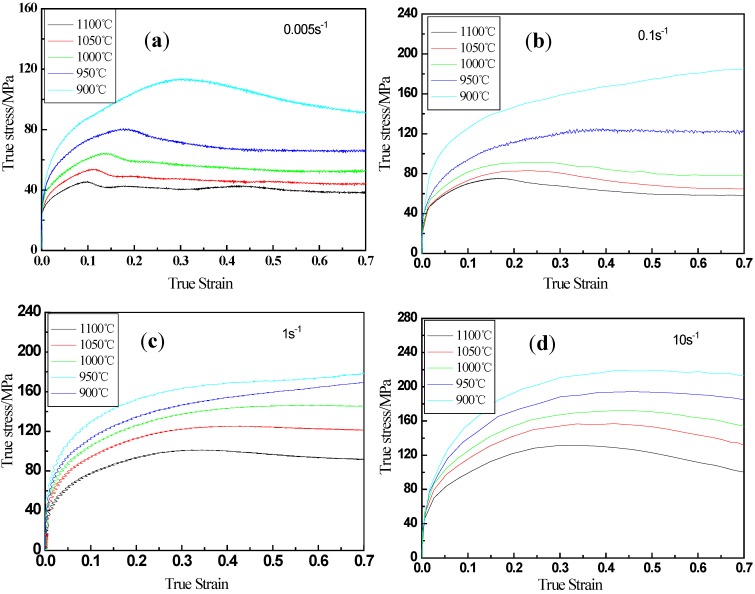
Flow curves of the examined steel at different temperatures and strain rates of (**a**) 0.005 s^−1^, (**b**) 0.1 s^−1^, (**c**) 1 s^−1^ and (**d**) 10 s^−1^.

**Figure 2 materials-13-01501-f002:**
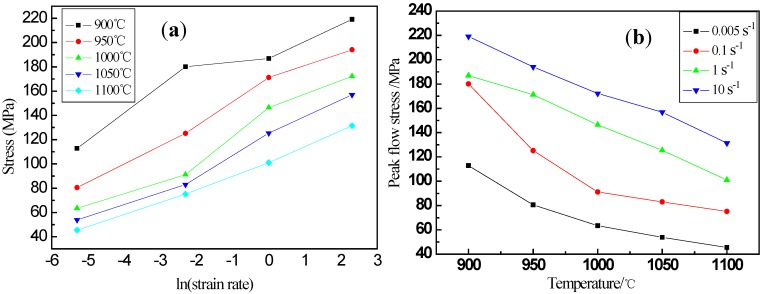
Relation between peak flow stress with (**a**) ln (strain rate) and (**b**) deformation temperatures.

**Figure 3 materials-13-01501-f003:**
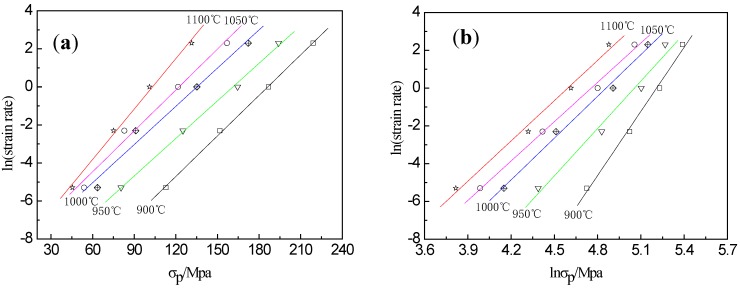
Relations between (**a**) ln(strain rate) and ln*σ_p_*, (**b**) ln(strain rate) and *σ_p_*, and (**c**) ln(strain rate) and ln[sinh(*ασ_p_*)] at different temperatures.

**Figure 4 materials-13-01501-f004:**
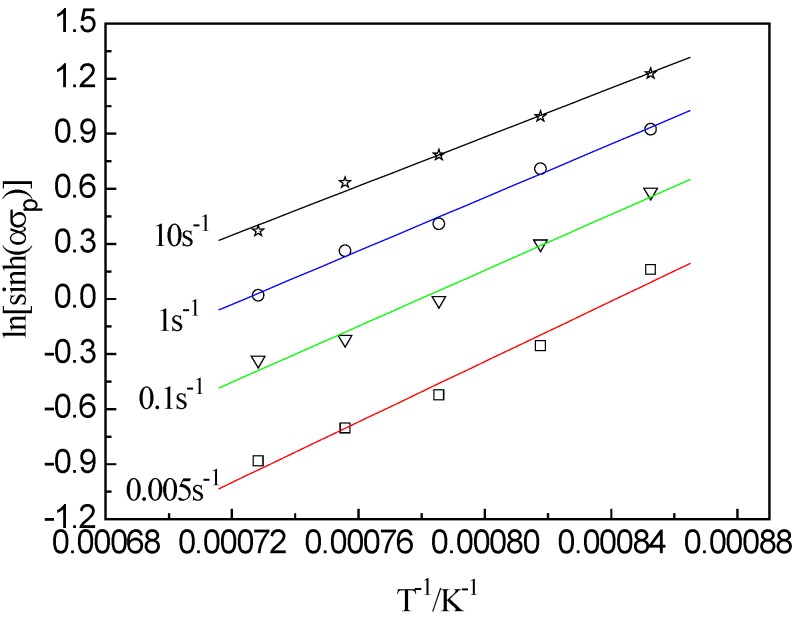
Relation between ln[sinh(*ασ_p_*)] and 1/*T* at different strain rate.

**Figure 5 materials-13-01501-f005:**
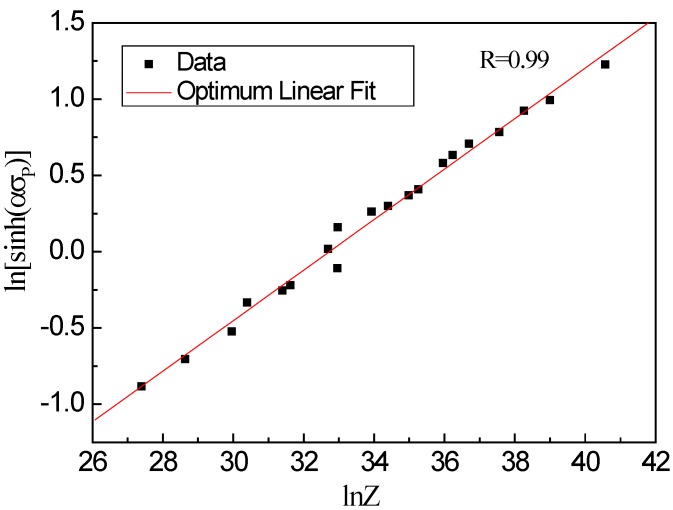
Relation between ln[sinh(*ασ_p_*)] and ln*Z*.

**Figure 6 materials-13-01501-f006:**
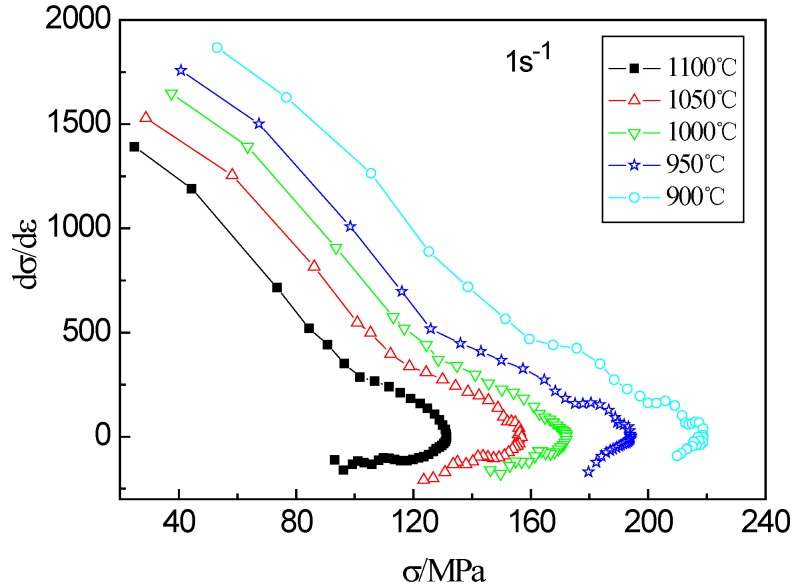
Strain hardening curves of the examined steel at 1 s^−1^ and different temperatures.

**Figure 7 materials-13-01501-f007:**
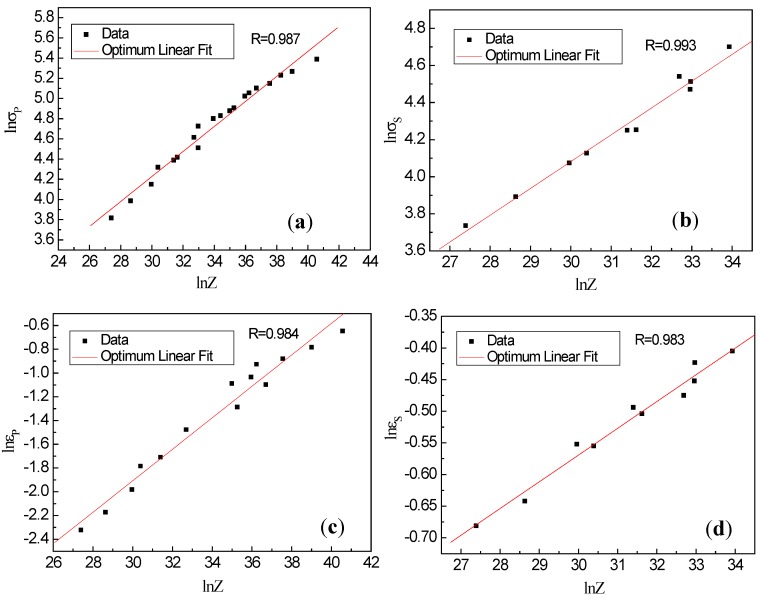
Relations between (**a**) peak stress and *Z*, (**b**) steady-state stress and *Z*, (**c**) peak strain and *Z* and (**d**) steady-state strain and *Z*.

**Figure 8 materials-13-01501-f008:**
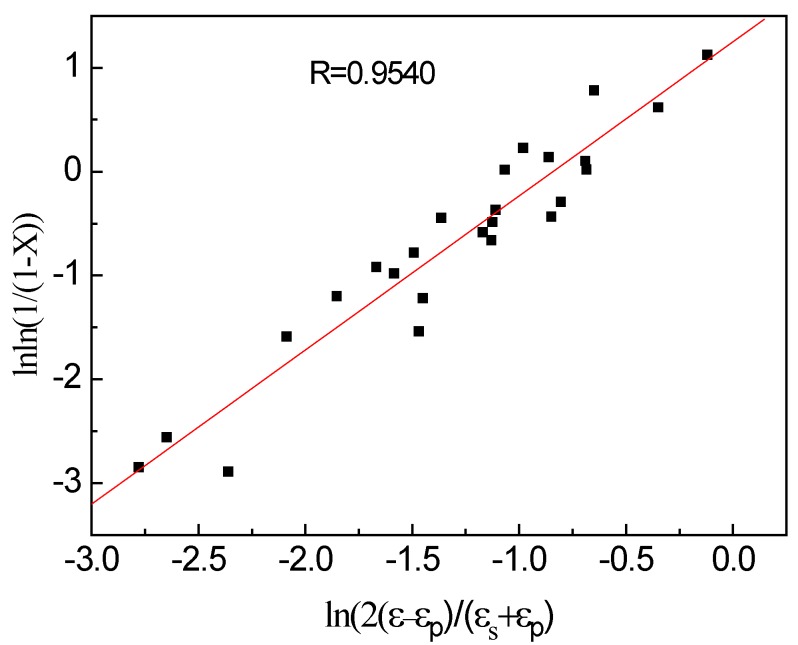
Relation between lnln(1/(1 − *X*)) and ln(2(*ε* − *ε_p_*)/(*ε_s_* + *ε_p_*)).

**Figure 9 materials-13-01501-f009:**
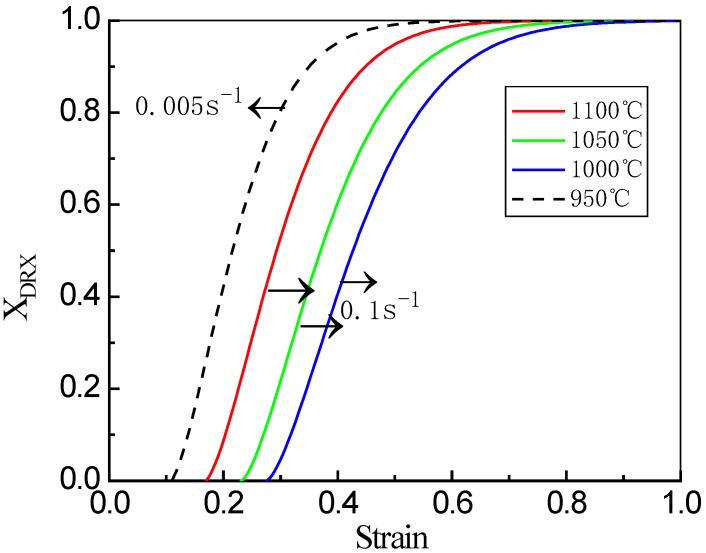
DRX kinetics curves for the examined steel at different temperatures and strain rates.

**Figure 10 materials-13-01501-f010:**
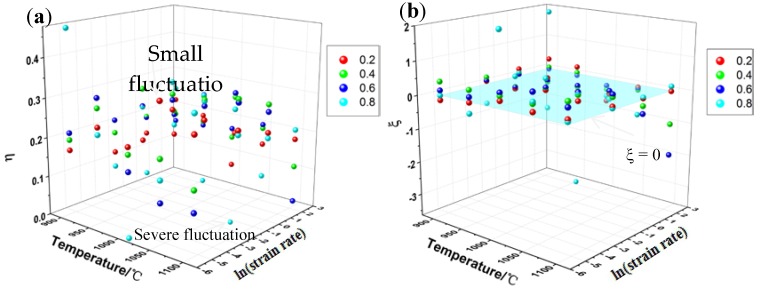
Three-D atomic distribution maps of value of (**a**) power dissipation efficiency and (**b**) instability parameter under different conditions.

**Figure 11 materials-13-01501-f011:**
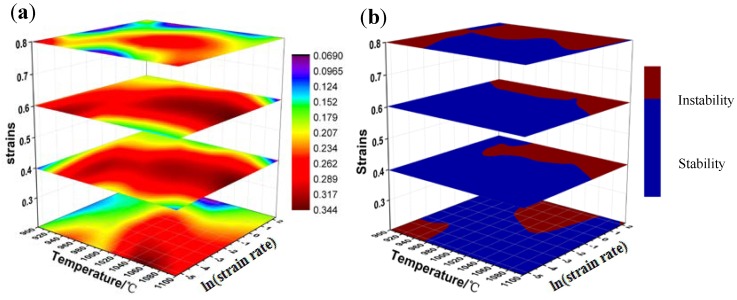
Three-D distribution maps of (**a**) power dissipation efficiency and (**b**) instability parameter under different conditions.

**Figure 12 materials-13-01501-f012:**
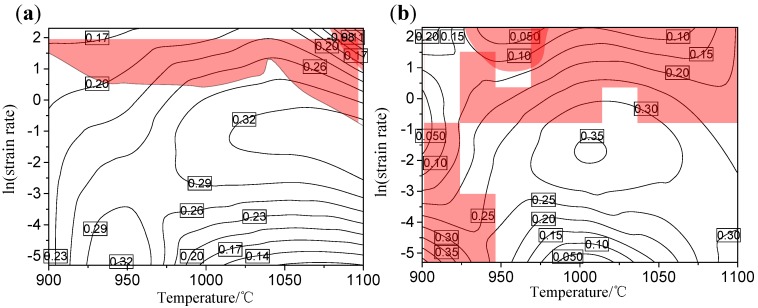
Processing maps of a high-titanium Nb-micro-alloyed steel obtained by superposition of the PDM and the instability map under true strains: (**a**) 0.6 and (**b**) 0.8.

**Figure 13 materials-13-01501-f013:**
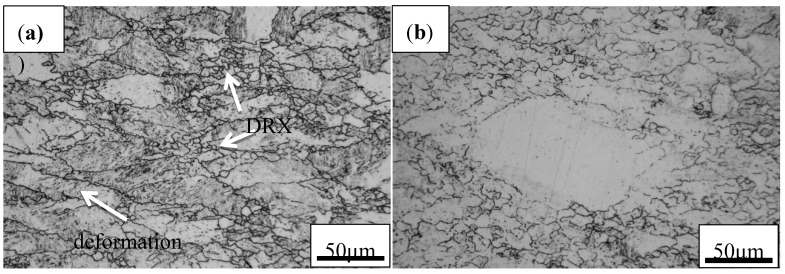
LM micrographs of the examined steel under deformation conditions of (**a**) 1000 °C and 0.005 s^−1^ and (**b**) 1050 °C and 10 s^−1^.

**Figure 14 materials-13-01501-f014:**
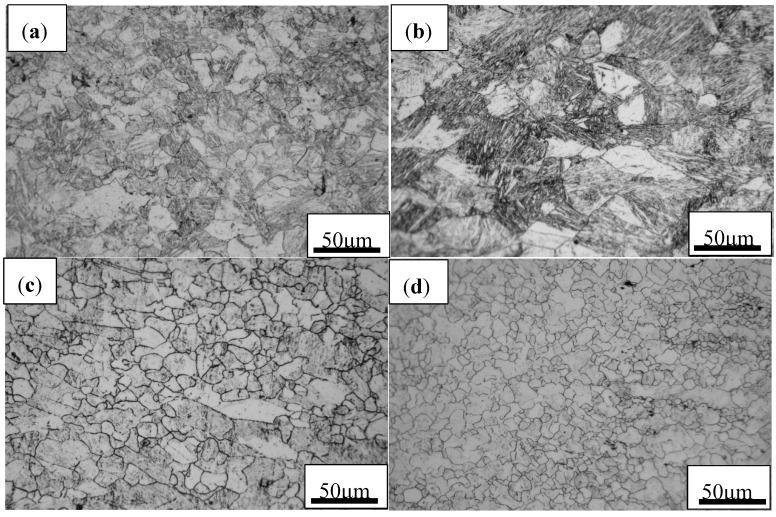
LM micrographs of the examined steel after hot deformation under different conditions: (**a**) 950 °C and 0.005 s^−1^, (**b**) 950 °C and 10 s^−1^, (**c**) 1050 °C and 0.1 s^−1^ and (**d**) 1000 °C and 0.1 s^−1^.

**Figure 15 materials-13-01501-f015:**
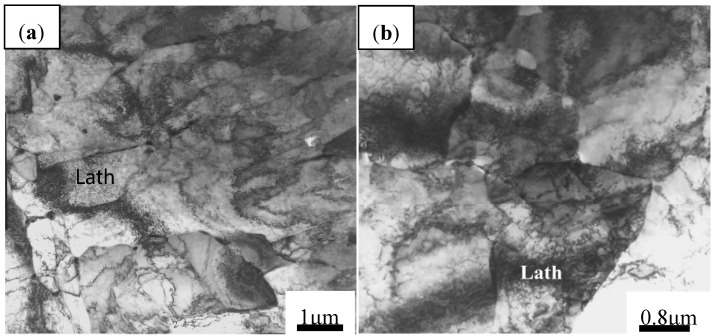
The TEM micrographs of Nb–Ti micro-alloyed steel at different conditions: (**a**) 1050 °C and 0.1 s^−1^ and (**b**) 1050 °C and 10 s^−1^.

**Table 1 materials-13-01501-t001:** Chemical composition of the examined steel (wt %).

Elements	C	Mn	Ti	Nb	Si	N	S	Al	Cr	Fe
Composition	0.093	1.54	0.104	0.078	0.3	0.0051	0.0056	0.013	0.43	Bal.

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
