# Peer review of "Hot Deformation Characteristics and 3-D Processing Map of a High-Titanium Nb-Micro-alloyed Steel"

_materials, 2020, doi:10.3390/ma13071501_

Round 1

Reviewer 1 Report

In the caption of the Figure 2 should be written that in the y-axis are presented the values of peak flow stress. In Figure 2a the markers on the x-axis are missing. On the line 107 is listed the wrong unit of temperature T(k) - the correct should be T (K). In which place was detected in the Figure 1 the value of peak flow stress in the true stress - true strain curve for deformations conditions strain rate 0,1 1/s and temperature 900 °C? From the presented stress - strain curve it can not be clearly determined the peak flow stress. In which place was detected in the Figure 1 the value of peak flow stress in the true stress - true strain curves for strain rate 1 1/s and temperature 900 °C or 950 °C? The values of peak flow stress determined from true stress - true strain curves (in Figure 1) for these conditions do not agree with values presented in Figure 2. The units presented on the axes are presented in different format. For example on the axes of Figure 2 are the units in the round brackets, but in the more cases (Figure 3, 4, 6, etc.) the round brackets are not used. In the Figure 4 are used at description of individual curves the wrong units of strain rate 1/S (with a capital letter S) - the unit of strain rate is 1/s (with lowercase letter). In the Figure 7b is wrong description of the axis y. The correct desription shouold be lnσss is flow stress for steady state). Line 162 - in this line s erroneously referenced to equation (Eq. 5) - the correct should be equation (Eq. 10). In the Figure 11b is wrong description of the scale. The area of stability should be in blue color and the area of instability should be in red color. In the Figure 13 and Figure 15 the scale is not sufficiently legible.

Author Response

Response to Reviewer 1 Comments

Point 1: In the caption of the Figure 2 should be written that in the y-axis are presented the values of peak flow stress. In Figure 2a the markers on the x-axis are missing. On the line 107 is listed the wrong unit of temperature T(k) - the correct should be T (K).

Response 1: This problem has been solved in Figure 2. On line 110, the problem has been modified.

Point 2: In which place was detected in the Figure 1 the value of peak flow stress in the true stress - true strain curve for deformations conditions strain rate 0.1 s-1 and temperature 900℃? From the presented stress - strain curve it can not be clearly determined the peak flow stress. In which place was detected in the Figure 1 the value of peak flow stress in the true stress - true strain curves for strain rate 1 s-1 and temperature 900℃ or 950℃? The values of peak flow stress determined from true stress - true strain curves (in Figure 1) for these conditions do not agree with values presented in Figure 2.

Response 2: It can be seen from Figure 1 and experimental data obtained by thermal simulation computer that at strain rate 0.1 s-1 and temperature 900℃, the peak stress appears near strain 0.668 and is 180.0379 MPa. The peak stress in Figure 1 appears near the maximum strain for strain rate 1 s-1 and temperature 900℃ or 950℃, which is not obviously detected. After inspection, the peak flow stress value determined by the true stress true - strain curve (Figure 1) is consistent with the value shown in Figure 2.

The peak stress at different deformation temperature and strain rate:

      0.005      0.1        1        10

900   112.8624  180.0379  186.8714   219.08

950   80.5488 125.13  171.1581 194.13

1000  63.4966 91.1392  146.4277 172.17

1050  53.8532 82.9724  125.3981 156.89

1100  45.4533 75.1006  101.0212 131.41

Point 3: The units presented on the axes are presented in different format. For example on the axes of Figure 2 are the units in the round brackets, but in the more cases (Figure 3, 4, 6, etc.) the round brackets are not used. In the Figure 4 are used at description of individual curves the wrong units of strain rate 1/S (with a capital letter S) - the unit of strain rate is 1/s (with lowercase letter). In the Figure 7b is wrong description of the axis y. The correct desription shouold be lnσs (σs is flow stress for steady state). Line 162 - in this line s erroneously referenced to equation (Eq. 5) - the correct should be equation (Eq. 10). In the Figure 11b is wrong description of the scale. The area of stability should be in blue color and the area of instability should be in red color. In the Figure 13 and Figure 15 the scale is not sufficiently legible.

Response 3: For all graphs, slash symbols have been used instead of round brackets. In the Figure 4 the problem has been modified. In the Figure 7b the problem has been modified. On line 164, this problem has been solved and referenced to the correct equation (Eq. 10). In the Figure 11b this problem has been modified. In the Figure 13 and Figure 15 the scale has been modified to normal.

The conclusion 1, 2 and 4 have been greatly modified.

Reviewer 2 Report

Lines 45 – 47: The listed references [10 – 20] which are dealing with the processing maps should be enlarged about some newer articles, check for example https://www.mdpi.com/.

Line 117: the round bracket is missing in the expression ln[sinh(…….

Line 162: There is a link to Eq. 5. However, there should be link to Eq. 10.

Figure 7b.: The y-axis label is not correct. There should be subscript “s” instead of epsilon.

Figure 11b: The colors in the color bar are labeled conversely, i.e. the red part should be labeled as instability and the blue one as stability (in accordance with the text in the line 231).

Line 234: the strain rates 0.005-0.1 s-1 and 1-10 s-1 are not corresponding with the instability area at the strain of 0.6 in Fig. 11b. This is probably because of the above-mentioned color confusion.

Figure 13: In the caption of this figure, there should be the term “true strain” instead of “ture strain”.

Additional comments:

Figures 13 and 15: There should be stated unit micrometer in the scale instead of only micro.

In the submitted manuscript, there is often used the natural-logarithm symbol (ln). However, in many cases (text and even figures), the small letter l (“el”) is replaced by the capital letter I (“aye”) – this should be corrected.

Author Response

Response to Reviewer 2 Comments

Point 1: Lines 45 – 47: The listed references [10–20] which are dealing with the processing maps should be enlarged about some newer articles, check for example https://www.mdpi.com/. 

Response 1: The latest articles about processing maps have been added, which are [15], [21], [22].

Point 2: Line 117: the round bracket is missing in the expression ln[sinh(…….

Response 2: Missing round bracket is added to expression on page 120.

Point 3: Line 162: There is a link to Eq. 5. However, there should be link to Eq. 10.

Response 3: Line 164 - in this line s erroneously referenced to equation (Eq. 5) - the correct should be equation (Eq. 10).

Point 4: Figure 7b.: The y-axis label is not correct. There should be subscript “s” instead of epsilon.

Response 4: In Figure 7b, the y-axis label has been modified to the subscript "s".

Point 5: Figure 11b: The colors in the color bar are labeled conversely, i.e. the red part should be labeled as instability and the blue one as stability (in accordance with the text in the line 231).

Response 5: In Figure 11b, the color mark in the color bar has been modified.

Point 6: Line 234: the strain rates 0.005-0.1 s-1 and 1-10 s-1 are not corresponding with the instability area at the strain of 0.6 in Fig. 11b. This is probably because of the above-mentioned color confusion.

Response 6: In the line 237-238, the strain of 0.6 of instability area has been modified to 900-1100 ℃/1-10 s-1.

Point 7: Figure 13: In the caption of this figure, there should be the term “true strain” instead of “ture strain”.

Response 7: In the caption of this figure correct the "true strain" writing.

Point 8: Additional comments: Figures 13 and 15: There should be stated unit micrometer in the scale instead of only micro.

In the submitted manuscript, there is often used the natural-logarithm symbol (ln). However, in many cases (text and even figures), the small letter l (“el”) is replaced by the capital letter I (“aye”) – this should be corrected.

Response 8: In the Figure 13 and 15 the scale has been modified to normal. In the  text, the small letter l (“el”) has been replaced by the capital letter I (“aye”) in the paper.

Reviewer 3 Report

Paper deals with useful topic of hot deformation of HSLA steel based on Ti-Nb.

It is interesting, however significant improvements are needed. Some of issues are listed below:

  1. English language needs extensive editing to make it readable easily.
  2. For processing maps review in introduction author stated is not first author.
  3. In experimental methods full sample prep technique is needed.
  4. Discussion of true strain-true stress curves is very speculative regarding DRV component of softening.
  5. At strain rate 10 s^{-1} steady state is not reached, but in strain hardening curves in fig. 6 it is implied that it is reached. 
  6. In equation 10 it is not explained how k and n3 were obtained.
  7. In conclusions it is suggested optimal temperature is 1000 °C and strain rate suggested is below what is usually done in rolling mill. Can you discuss something why this temperature range is optimal? NbC and Nb(C,N) are expected to precipitate at theses condition which have influence on DRX. 

Author Response

Response to Reviewer 3 Comments

Point 1: English language needs extensive editing to make it readable easily.

Response 1: The language has been properly edited and improved.

Point 2: For processing maps review in introduction author stated is not first author.

Response 2: The expression of this sentence is not accurate. On page 38-40, it was modified to "It was proposed by Raj et al. to build a process map on the basis of cavity nucleation mechanisms, dynamic recrystallization and adiabatic heating effects".

Point 3: In experimental methods full sample prep technique is needed.

Response 3: On page 73-79, in the experimental method the sample preparation technique and process are supplemented in detail.

Point 4: Discussion of true strain-true stress curves is very speculative regarding DRV component of softening.

Response 4: You are right, it is difficult to obtain the softened DRV component from the true strain-true stress curve, so the sentence was deleted in the part of the analysis of the true strain-true stress curve, which is “it reveals that the conditions of deformation temperature and strain rate of the experimental steel can significantly affect the occurrence of DRX and DRV”.

Point 5: At strain rate 10 s-1 steady state is not reached, but in strain hardening curves in fig. 6 it is implied that it is reached. 

Response 5: This is a mark error in drawing, it is actually a strain rate of 1 s-1 steady state, which has been revised in the paper.

Point 6: In equation 10 it is not explained how k and n3 were obtained.

Response 6: On pages 166-167, it is explained how k and n3 were obtained, which is expressed as “The slope and intercept of the fitted linear equation in Figure 8 correspond to the values of the material coefficients k and n3, that is, they are 1.4847 and 3.4931, respectively.”

Point 7: In conclusions it is suggested optimal temperature is 1000 °C and strain rate suggested is below what is usually done in rolling mill. Can you discuss something why this temperature range is optimal? NbC and Nb(C,N) are expected to precipitate at theses condition which have influence on DRX. 

Response 7: On the one hand, according to the microstructure evolution, as shown in Fig. 14, the finer grain size can be obtained at 1000 ℃/0.1 s-1; on the other hand, according to the processing maps, as shown in Fig. 12 and Fig. 13, a medium-low strain rate at 1000 ℃ is conducive to better processability (higher power consumption efficiency value) and avoid the generation of processing defects. In order to obtain greater deformation stress (deformation resistance), rolling mills generally increase the strain rate to greatly increase the deformation resistance of the material, but ignore the high strain rate, which is prone to microcracks and other defects during rolling. The lower strain rate is beneficial to the dynamic recrystallization. Nb elements are usually precipitated as carbonitrides, making it difficult to form pure NbN or NbC. The precipitation temperature range of niobium carbonitride is generally in austenite area, its main function is to prevent the recrystallization of austenite, improve the recrystallization temperature of austenite and refine austenite grains.

The conclusion 1, 2 and 4 have been greatly modified.

Reviewer 4 Report

Submitted paper is quite interesting and quite well prepared. However some corrections are required (listed below).

page 2, line 58 – It is not necessary to place [wt%] unit twice. I suggest to leave in the table. Moreover, “experimental steel (or material)” refers to the material which is developed and/or modified in presented research. I suggest to use – in whole text – the term “examined steel (or material)”.

page 2, line 78 and further – I suggest to avoid a formulations like “Fig. 1 shows …”, “As shown in Fig. …” The sentence should unequivocally inform the reader, even graphical materials are not visible. For instance, “The true stress-true strain curves (NOT ONLY ONE!) of the experimental steel at different deformation temperatures for a given strain rate were analyzed (Fig. 1). When the strain rate is 0.1 s-1 and the deformation temperature is higher than 950 °C (Fig. 1b), the flow …”

page 4, between lines 117 and 118 – Shouldn’t be “ln strain rate” instead of “log strain rate” in the Y axis label?

page 6, lines 148, 149 – Fig. 7a presents relationship between peak stress and Z – or ln peak stress and lnZ – you should be consistent. This remark refers to Figs 7b,c,d as well.

page 12, lines 282 and 285 – Every micrograph is optical! You should write “LM micrographs …” and add to chapter 2. EXAMINED material and procedures” the methodology of light microscope (LM) observations and metallographic procedure.

page 12, line 289 – It is not necessary to write “Thin foil TEM micrographs …” – “TEM micrographs …” is enough. It is widely known what TEM specimen is.

Moreover, the unit in scale bars should be corrected.

page 13, line 311 – The conclusion no 2 is incomprehensible. Do you mean “The higher titanium and niobium CONTENT in the EXAMINED steel LEADS TO higher hot deformation ….”?

Author Response

Response to Reviewer 4 Comments

Point 1: page 2, line 58 – It is not necessary to place [wt%] unit twice. I suggest to leave in the table. Moreover, “experimental steel (or material)” refers to the material which is developed and/or modified in presented research. I suggest to use – in whole text – the term “examined steel (or material)”.

Response 1: The wt% in the table has been removed. The "examined steel (or material)" has been used instead of "experimental steel (or material)" in whole text.

Point 2: page 2, line 78 and further – I suggest to avoid a formulations like “Fig. 1 shows …”, “As shown in Fig. …” The sentence should unequivocally inform the reader, even graphical materials are not visible. For instance, “The true stress-true strain curves (NOT ONLY ONE!) of the experimental steel at different deformation temperatures for a given strain rate were analyzed (Fig. 1). When the strain rate is 0.1 s-1 and the deformation temperature is higher than 950 °C (Fig. 1b), the flow …”

Response 2: Similar expressions have been modified in the text, for example on line 83-87.

Point 3: page 4, between lines 117 and 118 – Shouldn’t be “ln strain rate” instead of “log strain rate” in the Y axis label?

Response 3: On page 4, the y-axis label of Fig. 3 has been modified to “In strain rate”.

Point 4: page 6, lines 148, 149 – Fig. 7a presents relationship between peak stress and Z – or ln peak stress and lnZ – you should be consistent. This remark refers to Figs 7b,c,d as well.

Response 4: page 6, lines 120, 151 – The title of Fig. 7 has been modified to show the relationship between peak stress and Z.

Point 5: page 12, lines 282 and 285 – Every micrograph is optical! You should write “LM micrographs …” and add to chapter 2. EXAMINED material and procedures” the methodology of light microscope (LM) observations and metallographic procedure.

Response 5: page 12, lines 286 and 289 – “Optical micrographs” has been revised to “LM micrographs” and material and procedures and others.

Point 6: page 12, line 289 – It is not necessary to write “Thin foil TEM micrographs …” – “TEM micrographs …” is enough. It is widely known what TEM specimen is.

Moreover, the unit in scale bars should be corrected.

Response 6: page 12, line 292 – “Thin foil TEM micrographs …” has been revised to “TEM micrographs …”. In the Figure 13 and Figure 15 the scale has been modified to normal.

Point 7: page 13, line 311 – The conclusion no 2 is incomprehensible. Do you mean “The higher titanium and niobium CONTENT in the EXAMINED steel LEADS TO higher hot deformation ….”?

Response 7: Yes, it has been modified that “The higher titanium and niobium content in the examined steel could leads to higher the activation energy of hot deformation and impeded the occurrence of dynamic recrystallization”. The conclusion 1, 2 and 4 have been greatly modified.

Round 2

Reviewer 3 Report

Some comments were followed in revision.

It is stil difficult to read the paper, as the explanations are too confusing and the style used does not allow us to understand the explanations given. For example; "The microstructure observations were carried out along a cut along parallel to the compression axis, then the surfaces were etched in a solution containing of 15 ml picric acid, 200 ml distilled
75 water and 3 drops of hydrofluoric acid to reveal the prior austenite grain structure." - this sentence is so confusing and scientifically wrong.

Please explain 3 drops stated in SI units.

Response 2: there is missing reference for Gegel et al. in line 42

Please rewrite conclusions so reader can understand them. For example, "The higher titanium and niobium content in the examined steel could leads to higher the activation energy of hot deformation and impeded the occurrence of dynamic recrystallization..." here it is impossible to understand what is being said.

Conclusion 1 is nothing especial. Most materials behave in this way as at low strain rate and high temperatures driving force for DRX is higher. Diffusion is easier and time is longer.

Author Response

Point 1: It is stil difficult to read the paper, as the explanations are too confusing and the style used does not allow us to understand the explanations given. For example; "The microstructure observations were carried out along a cut along parallel to the compression axis, then the surfaces were etched in a solution containing of 15 ml picric acid, 200 ml distilled water and 3 drops of hydrofluoric acid to reveal the prior austenite grain structure." - this sentence is so confusing and scientifically wrong.

Please explain 3 drops stated in SI units.

Response 1: This sentence has modified its expression and scientifically wrong, that is “The deformed specimens were sectioned parallel to the compression axis for microstructural observation. The specimens were mechanical polished and etched in an abluent solution of saturated picric acid (15 ml picric acid, 200 ml water and 0.15 ml of hydrofluoric acid).” The 3 drops represent 0.15ml in SI units. Some languages have been appropriately improved. For example: in line 47-49; 74-81. etc.

Point 2: Response 2: there is missing reference for Gegel et al. in line 42

Response 2: The reference [9] represents reference for Gegel et al. in line 44.

Point 3: Please rewrite conclusions so reader can understand them. For example, "The higher titanium and niobium content in the examined steel could leads to higher the activation energy of hot deformation and impeded the occurrence of dynamic recrystallization..." here it is impossible to understand what is being said.

Conclusion 1 is nothing especial. Most materials behave in this way as at low strain rate and high temperatures driving force for DRX is higher. Diffusion is easier and time is longer.

Response 3: This sentence was revised at the request of another reviewer. Perhaps due to the problem of language expression, the expression of this sentence has been revised. Conclusions have been rewritten to understand them.